# Availability of the HPV Vaccine in Regional Pharmacies and Provider Perceptions Regarding HPV Vaccination in the Pharmacy Setting

**DOI:** 10.3390/vaccines10030351

**Published:** 2022-02-24

**Authors:** Jill M. Maples, Nikki B. Zite, Oluwafemifola Oyedeji, Shauntá M. Chamberlin, Alicia M. Mastronardi, Samantha Gregory, Justin D. Gatwood, Kenneth C. Hohmeier, Mary E. Booker, Jamie D. Perry, Heather K. Moss, Larry C. Kilgore

**Affiliations:** 1Department of Obstetrics and Gynecology, Graduate School of Medicine, University of Tennessee, Knoxville, TN 37920, USA; nzite@utmck.edu (N.B.Z.); ammastronardi@utmck.edu (A.M.M.); sgregory1@utmck.edu (S.G.); mbooker@utmck.edu (M.E.B.); jperry@utmck.edu (J.D.P.); hmoss@utmck.edu (H.K.M.); lkilgore@utmck.edu (L.C.K.); 2Department of Public Health, University of Tennessee, Knoxville, TN 37996, USA; oonaade@vols.utk.edu; 3Department of Family Medicine, Graduate School of Medicine, University of Tennessee, Knoxville, TN 37920, USA; schamberlin@utmck.edu; 4Department of Clinical Pharmacy and Translational Science, College of Pharmacy, University of Tennessee Health Science Center, Nashville, TN 37211, USA; jgatwood@uthsc.edu (J.D.G.); khohmeier@uthsc.edu (K.C.H.)

**Keywords:** HPV, HPV vaccine, pharmacy, vaccine referral, family medicine, obstetrics and gynecology, HPV vaccine barriers, provider perspectives

## Abstract

There is increasing support for HPV vaccination in the pharmacy setting, but the availability of the HPV vaccine is not well known. Additionally, little is known about perceptions of medical providers regarding referring patients to community pharmacies for HPV vaccination. The purpose of this study was to determine HPV vaccine availability in community pharmacies and to understand, among family medicine and obstetrics–gynecology providers, the willingness of and perceived barriers to referring patients for HPV vaccination in a pharmacy setting. HPV vaccine availability data were collected from pharmacies in a southern region of the United States. Family medicine and obstetrics–gynecology providers were surveyed regarding vaccine referral practices and perceived barriers to HPV vaccination in a community pharmacy. Results indicated the HPV vaccine was available in most pharmacies. Providers were willing to refer patients to a community pharmacy for HPV vaccination, despite this not being a common practice, likely due to numerous barriers reported. Pharmacist-administered HPV vaccination continues to be a commonly reported strategy for increasing HPV vaccination coverage. However, coordinated efforts to increase collaboration among vaccinators in different settings and to overcome systematic and legislative barriers to increasing HPV vaccination rates are still needed.

## 1. Introduction

Human papillomavirus (HPV) is the leading cause of most anogenital (i.e., anal, vulvar, vaginal, cervical, penile) and oropharyngeal cancers in both men and women, and the number of cancer cases linked to HPV has increased significantly over the past 15 years [1,2,3]. HPV-related cancers are the only cancers that can be prevented by a highly effective and safe vaccine [1,3]. In 2020, 75.1% of adolescents received at least one dose of an HPV vaccine, and 58.6% of adolescents were up to date with the entire series [4]. However, recent evidence suggests the COVID-19 pandemic resulted in fewer adolescents initiating the HPV vaccine series [4]. Among adults, the rates of HPV vaccination have increased moderately among certain populations over the past decade. For example, HPV vaccination coverage among males aged 19–26 years and Hispanic females aged 19–26 years increased, but approximately 50% of females aged 19–26 years and 70% of males aged 19–26 years remained unvaccinated [5]. Disparities in HPV vaccination coverage and HPV-cancer incidence exist in geographic locations in the United States [6,7,8]. Specifically, those living in rural areas [8] and in the Southern United States [6,7] have lower HPV vaccination rates and higher HPV-related cancer incidence rates.

To address vaccination shortfalls, the President’s Cancer Panel (2018) [9] and National Vaccine Advisory Committee (2016) [10] released statements supporting HPV vaccination utilizing community pharmacies. Pharmacists are ideally positioned to overcome some of the barriers to HPV vaccination initiation and series completion [11]. HPV vaccine administered in a pharmacy is a frequently mentioned strategy for increasing HPV vaccination coverage. Pharmacies are conveniently located for many families with easier access, especially in rural communities. In fact, most U.S. residents (91%) live within five miles of a community pharmacy [12]. Additionally, compared to doctors’ offices, pharmacies are open for longer hours and on weekends, which potentially facilitates improved access to vaccines [13]. Pharmacists in most states are authorized to administer vaccines and provide a convenient option for patients [14]. The extent to which pharmacies stock and administer the HPV vaccine, particularly in rural settings, is unknown.

While pharmacies could potentially serve a powerful role in providing HPV vaccine access, administration, and series completion, relatively little is known about the willingness of physicians to refer patients to a pharmacy setting for HPV vaccination. Campos-Outcalt et al. (2010) reported that 34.2% of family physicians referred adult patients to a pharmacy for routinely recommended vaccines, but the authors did not report if they would refer adolescents and/or adult patients to a pharmacy setting for HPV vaccination specifically [15]. A recent study by O’Leary et al. (2020) reported that among obstetrician–gynecologists that routinely assess for patient vaccination status, 92% screened for HPV vaccination status [16]. However, not all of these providers stocked and/or administered the HPV vaccine [16]. For obstetrician–gynecologists that are unable to stock and administer vaccinations onsite, the American College of Obstetrics and Gynecology (ACOG) has created guidance to develop immunization referral systems [17]. This guidance was developed with the goal of future pharmacy–physician practice collaboration. However, to our knowledge, there are no studies reporting the willingness of and perceived barriers to referring patients for HPV vaccination in a pharmacy setting among family medicine and obstetrics–gynecology providers.

The purpose of this study was to determine HPV vaccine availability in community pharmacy settings in a southern region of the United States. Additionally, this study aimed to understand, among family medicine and obstetrics–gynecology providers, the willingness of and perceived barriers to referring patients for HPV vaccination in a community pharmacy setting. This was accomplished by surveying family medicine and obstetrics–gynecology providers at a single academic medical center in the same geographic region.

## 2. Materials and Methods

### 2.1. Pharmacy Data Collection

The availability of the HPV vaccine in the pharmacy setting was determined by the percentage of pharmacies that administer and stock the HPV vaccine. To accomplish this, a list of all pharmacies in a southern region of the United States (consisting of 17 counties in Eastern Tennessee) was generated. First, a list of zip codes for the target counties was generated and then an online search for pharmacies located in those zip codes was performed. Each pharmacy was called by a research team member acting as a “secret shopper”, where the team member acted as a customer inquiring about pharmacy services from October 2020 through April 2021. A standardized script for data collection was developed primarily based on study outcomes, including HPV vaccine availability and other measures of accessibility (if the pharmacy stocked the HPV vaccine, how soon one could get the HPV vaccine, and if an appointment was needed to get the HPV vaccine). The script was discussed and confirmed by consensus among coauthors (J.M.M., O.O., S.G., J.D.G., K.C.H.). Using this standardized script, data regarding HPV vaccine availability were collected. The pharmacy staff member was first asked if the pharmacy administered the HPV vaccine. If the pharmacy stocked the HPV vaccine, they were then asked how soon one could get the HPV vaccine at their location, and if an appointment was required. Pharmacies were deemed “successfully contacted” if a research team member was able to call the pharmacy and get a definitive answer to any of the HPV vaccine-related questions. All pharmacies were considered in analyses. Up to three attempts were made to contact each pharmacy.

To better understand the impact of pharmacy type on HPV vaccine availability, pharmacies were categorized into the following groups: single independent, multiple independent, regional chain, grocery store chain (e.g., Kroger, Publix), national chain (e.g., Walgreens, CVS), and mass merchandiser (e.g., Wal-Mart, Target). To interpret the impact of pharmacy geographic location, categorizations of rural versus nonrural were created for each pharmacy location’s zip code based on Health Resources and Services Administration’s Federal Office of Rural Health Policy (HRSA FORHP) designations [18].

### 2.2. Family Medicine and Obstetrics–Gynecology Provider Surveys

An electronic survey was adapted from a previously published survey on adult immunization and preventative care practices by Hurley et al. [19]. The survey asked providers if their practice administered and stocked the HPV vaccine. Questions about Advisory Committee on Immunization Practices (ACIP)-recommended vaccine referral practices and then specifically about referral and prescription practices for the HPV vaccine across three different age categories for patients (age 11–18 years, age 19–26 years, and age 27–49 years) were included. Additionally, the survey asked about perceived barriers to HPV vaccination in a community pharmacy setting. The survey was emailed, via REDCap, to family medicine and obstetrics–gynecology providers (including physicians and midwives) at a single academic medical center. Research Electronic Data Capture (REDCap) is a secure, HIPAA-compliant, web application hosted by the University of Tennessee Graduate School of Medicine for building and managing online surveys and databases.

### 2.3. Statistical Analysis

Data analysis was primarily descriptive, using categorical data, means, and frequencies to determine vaccine availability and accessibility. Chi-squared analyses determined if HPV vaccine availability differed by pharmacy type and geographic location and if pharmacy type differed by geographic location. To determine HPV vaccine availability in the family medicine and obstetrics–gynecology providers’ practices and to understand HPV vaccine referral practices, survey data analysis was primarily descriptive, using categorical data, means, and frequencies. Data were analyzed using Microsoft Excel (Office 365) and SPSS (Released 2019. IBM SPSS Statistics for Windows, Version 26.0. IBM Corp, Armonk, NY, USA).

## 3. Results

### 3.1. Pharmacy Data Collection

A total of 233 out of 240 (97.1%) pharmacies were successfully contacted. Characteristics of the pharmacies contacted are presented in Table 1 along with measures of HPV vaccine accessibility and availability. Among the pharmacies contacted, the majority were national chain pharmacies (35.2%), followed by single independent pharmacies (27.5%). More than half of the pharmacies administered the HPV vaccine (60.1%), and among those that administered it, over two-thirds (67.1%) reported having it “in stock”. Ninety percent of the locations did not require an appointment to receive the vaccine, and 76.4% reported that the vaccine could be received the same day as the call of inquiry.

HPV vaccine availability differed significantly (*p* < 0.01) across pharmacy type (Figure 1). A larger percentage of mass merchandisers (24 out of 25, 96.0%), national chain pharmacies (76 out of 82, 92.7%), and grocery store pharmacies (23 out of 32, 71.9%) provide the HPV vaccine compared to single independent pharmacies (13 out of 64, 20.3%), multiple independent pharmacies (2 out of 24, 8.3%), and regional chain pharmacies (2 out of 6, 33.3%). HPV vaccine availability in pharmacies stratified by geographic location (rural vs. nonrural) was not significantly different (*p* = 0.704). Additionally, there were no significant differences in pharmacy type when stratified by geographic location (*p* = 0.704).

### 3.2. Family Medicine and Obstetrics–Gynecology Providers

#### 3.2.1. Provider Characteristics

The survey was distributed to 70 family medicine and Ob/Gyn providers; 60 providers completed the survey (85.7% response rate). Among the respondents, 34 (56.7%) were affiliated with the Department of Family Medicine at the University of Tennessee Graduate School of Medicine and 26 (43.3%) were affiliated with the Department of Obstetrics and Gynecology at the same institution. The majority identified as female (*n* = 38, 63.3%), and the age of the respondents ranged from 26 to 69 years of age (mean = 35.9 ± 10.6 years). In terms of role, 55.0% (*n* = 33) were resident physicians, 41.7% (*n* = 25) were attending physicians, and 3.3% (*n* = 2) were midwife practitioners.

#### 3.2.2. HPV Vaccination Availability and Referral Practices

Most respondents indicated their practice administered and stocked the HPV vaccine. Nearly all (98.3%, *n* = 59) respondents reported their practice administers the HPV vaccine, and 95.0% stock the vaccine. When asked if they had ever referred a patient to receive an ACIP-recommended vaccine outside of their practice, 81.7% (*n* = 49) reported they had referred a patient for vaccination. When asked specifically if they have ever referred a patient for HPV vaccination outside of their practice, 20.0% (*n* = 12) had referred a patient age 11–18 years, 21.7% (*n* = 13) had referred a patient age 19–26 years, and 20.0% (*n* = 12) had referred a patient age 27–49 years. Among those that had ever referred a patient for HPV vaccination outside of their practice, over 90.0% referred patients to a local health department for HPV vaccination and did so across all patient age groups. Regarding referrals to a community pharmacy for HPV vaccination, 50% of those ever referring a patient for HPV vaccination referred a patient age 27–49 years, 53.8% referred a patient age 19–26 years, and 33.3% referred a patient age 11–18 years.

When asked if they had ever prescribed the HPV vaccine for a patient to be received at a community pharmacy location, 6.7% (*n* = 4) reported they had prescribed the HPV vaccine for patients age 19–26 years, followed by 5.0% (*n* = 3) and 3.3% (*n* = 2) for patients age 27–49 and 11–18 years respectively. Most providers (>90%) were willing or very willing to refer an eligible patient to receive an HPV vaccination at a community pharmacy across all age groups (Table 2).

Perceived barriers to referring a patient for HPV vaccination in a pharmacy setting are shown in Figure 2. The most-reported barrier (61.7% of respondents) indicated that providers felt it made the most sense that their patients receive the HPV vaccine in their office (Figure 2).

When asked what percentage of pharmacies in their region administer and stock the HPV vaccine, 88.3% (*n* = 53) of providers answered, “I do not know”. Among the few respondents (11.7%, *n* = 7) that provided a numeric answer to the question regarding the percentage of pharmacies that administer the HPV vaccine, answers ranged from 25% to 85%, which was similar to the answers provided in response to the question of percentage of pharmacies that stock the HPV vaccine (ranging from 10% to 90%).

## 4. Discussion

The purpose of this study was to determine HPV vaccine accessibility in pharmacy settings in a southern region of the United States and to understand, among family medicine and obstetrics–gynecology providers, the willingness of and perceived barriers to referring patients to community pharmacies for HPV vaccination. The primary findings indicated that the majority of pharmacies in Eastern Tennessee (a state in the Southern United States) administer and stock the HPV vaccine. Family medicine and obstetrics–gynecology providers surveyed in this study were willing to refer patients to a community pharmacy for HPV vaccination, despite this not being a common practice. Most providers did not know the extent to which community pharmacies in their region administered and/or stocked the HPV vaccine. Providers also reported several perceived barriers to referring patients for HPV vaccination in a pharmacy setting. The most common barriers expressed included their desire for patients to be vaccinated in their office; concern that patients would not complete the HPV vaccine series if referred outside their practice for vaccination; providers not knowing or having documentation of vaccination if performed in a pharmacy setting; and potential financial burdens for patients, associated with lack of insurance coverage, for the HPV vaccine in a pharmacy setting.

In the current study, a larger proportion of pharmacies (60.1%) reported administering the HPV vaccine (Table 1) compared to previous reports [13,20]. A study by Hastings et al. (2017) reported that only 11.7% of community pharmacies in the state of Alabama, which is another state located in the Southern United States, had the HPV vaccine in their inventory [20]. A study by Westrick et al. (2018) reported that, among a nationally representative sample of pharmacies in the United States, 38.9% of pharmacies offered the HPV vaccine [13]. However, 44.2% of the sample consisted of independently owned pharmacies, and the percentage of pharmacies offering the HPV vaccine was not stratified by pharmacy type [13]. In the current study, mass merchandisers (96.0%), national chain pharmacies (92.7%), and grocery store pharmacies (71.9%) reported administering the HPV vaccine, while fewer single independent pharmacies (20.3%), multiple independent pharmacies (8.3%), and regional chain pharmacies (33.3%) offered the HPV vaccine (Figure 1). This is consistent with previous reports that describe barriers to HPV vaccination among independent pharmacies [21,22]. These include organizational barriers to HPV vaccine administration including lack of space and staff.

Reports indicate HPV vaccination coverage is lower in rural areas, despite these areas having higher rates of HPV-associated cancers [4,8]. In the current study, we surveyed pharmacies in a region where approximately 38% of the population lives in a rural area [23]. We found that HPV vaccine availability in pharmacy settings was not different between those in rural and nonrural locations. Additionally, there were no significant differences in pharmacy type when stratified by geographic location, which suggests that, in the region surveyed, the HPV vaccine is likely available in a chain, mass merchandiser, and/or grocery pharmacy setting even in rural locations. This is consistent with the concept that pharmacies in rural locations could potentially serve a powerful role in providing HPV vaccine administration [12,22]. However, the benefit of the HPV vaccine in the rural pharmacy setting alone is clearly not enough to increase HPV vaccination coverage as HPV vaccination uptake and accessibility is impacted by legislative, social, and environmental factors unique to HPV vaccination, nor would availability overcome other obstacles for vaccination including medical literacy, socioeconomics, and personal choice [11].

In the current study, family medicine and obstetrics–gynecology providers were willing to refer patients to a community pharmacy for HPV vaccination, despite this not being a common practice. A study by Hurley et al. (2014) reported on vaccination referral practices by alternate vaccinators, including general internists and family medicine providers, where financial barriers made them less likely to stock and administer vaccines [19]. Although the report was not specific to the HPV vaccine, the providers surveyed indicated they “always, often or sometimes” referred patients to a pharmacy for vaccination [19]. Additionally, the study reported that most providers agreed it was helpful to have pharmacists share a role in patient vaccination. However, one-third of providers had reservations about pharmacists as vaccinators. Some of these reservations were related to communication, or lack thereof, between physicians and pharmacists and concerns about inadequate documentation of patient vaccines. They disagreed that it was more convenient for their patients to get vaccines at a pharmacy compared to their office, and the majority reported their patients preferred to receive vaccines at their office, rather than a pharmacy. The results of the current study suggest similar themes presented in the Hurley et al. study. Providers support the idea of pharmacists as vaccinators (Table 2) but would prefer to vaccinate their patients in their offices (Figure 2). Perhaps this is tied to concern about fragmented care and the importance of patients receiving care in their medical home.

It is worthwhile to mention that among the providers surveyed in the current study, none of them perceived HPV vaccination as outside the scope of a pharmacist. This suggests the training and knowledge of a pharmacist surrounding the HPV vaccine may not be problematic for many family medicine and obstetrics–gynecology providers. Overall, these findings are in contrast to a study by Welch et al. (2003) that found most family medicine physicians were neither very knowledgeable about nor supportive of pharmacists as vaccinators [24]. It is possible the providers surveyed in the current study are inherently different compared to the providers surveyed in the Welch et al. study. For example, the providers included in the current study were from a single academic institution where the majority work in clinics that administer and stock the HPV vaccine. Similarly, it is possible that providers in clinical settings that are unable to or that chose not to administer or stock the HPV vaccine may have different HPV vaccine referral practices and/or attitudes towards referring patients to a pharmacy setting for HPV vaccination. In terms of generalizing the findings of the current study more broadly, several of the providers surveyed indicated they treat primarily adult patients, which could have influenced their willingness and comfort to refer patients to a pharmacy setting for HPV vaccination.

While the majority of pharmacies in the current study had the HPV vaccine available, the majority of providers in the current study were not actively referring patients to a pharmacy setting for HPV vaccination. Providers reported concerns their patients may not complete the HPV vaccine series if referred outside of their office for vaccination and expressed concern they would not be informed whether their patients received vaccines in a pharmacy setting (Figure 2). The lack of coordinated up-to-date immunization registries has been cited as a barrier in previous studies [21]. Promoting the wider use of immunization registries could be a strategy to overcome some provider concerns related to referring patients outside the office setting for vaccination.

In the current study, we did not specifically ask providers to distinguish their opinions and/or practices regarding HPV vaccine series initiation versus booster. For example, it is possible some providers prefer to administer the first HPV vaccine dose that a patient receives in their clinic, but would be willing to refer patients to a pharmacy setting for HPV vaccine series completion. Additional studies are needed to determine if providers might be more willing to refer patients to a pharmacy for HPV vaccine series completion and to determine the rate of follow-up for HPV vaccination among patients that are referred to a community pharmacy for HPV vaccination series completion. A recent study by Douchette et al. (2019) described a coordinated delivery of an HPV vaccine program using a clinic–pharmacy partnership. In their study, less than 50% of patients referred to a pharmacy received an HPV vaccine [25]. However, 100% of the patients that received the HPV vaccine in the pharmacy setting, after referral to the pharmacy for vaccination, completed the series. Further evidence of successful vaccine series completion in a community pharmacy setting is described in a recent report by Frederick et al. (2020). In this study, they describe the use of “nudge”-based clinical decision support embedded within a community pharmacy’s software system to improve vaccine series completion in a community pharmacy setting [26].

This study has limitations. These results may not be generalizable beyond this specific region in the Southern United States, as HPV vaccination uptake and accessibility are impacted by several factors that are state- and region-specific. These may include legislation and policies regarding HPV vaccination, social and environmental factors, religiosity, political ideology, and vaccine hesitancy. However, these data suggest that more research is needed to better understand perceived barriers and opportunities for HPV vaccination among providers in a variety of clinical settings. The providers included in the current study were from a single academic institution, working in clinics that administer and stock the HPV vaccine. It is possible that providers working in different settings would not be as willing to refer patients to a pharmacy setting for the HPV vaccine. Future work should include providers that are not affiliated with an academic medical center and those that do not work in practices that administer and stock the HPV vaccine. Finally, this study did not attempt to evaluate pharmacists as facilitators of HPV vaccines or their comfort in counseling patients about HPV vaccination. Nor did this study attempt to evaluate perceptions of facilitators to HPV vaccination in the pharmacy setting among family medicine and obstetrics–gynecology providers. Additional studies are needed to understand barriers and overcome obstacles to HPV vaccination, particularly in the community pharmacy setting.

## 5. Conclusions

Findings suggest that the HPV vaccine is commonly available in nonindependent pharmacies in Eastern Tennessee. Family medicine and obstetrics–gynecology providers report they are willing to refer patients for HPV vaccination in a pharmacy setting; however, several barriers were reported that might limit this practice. While pharmacist-administered HPV vaccination continues to be a commonly reported strategy for increasing HPV vaccination coverage and the availability of the HPV vaccine in the pharmacy setting is common, coordinated efforts to increase collaboration among vaccinators in different settings and overcome systematic and legislative barriers to increasing HPV vaccination rates are still needed.

## Figures and Tables

**Figure 1 vaccines-10-00351-f001:**
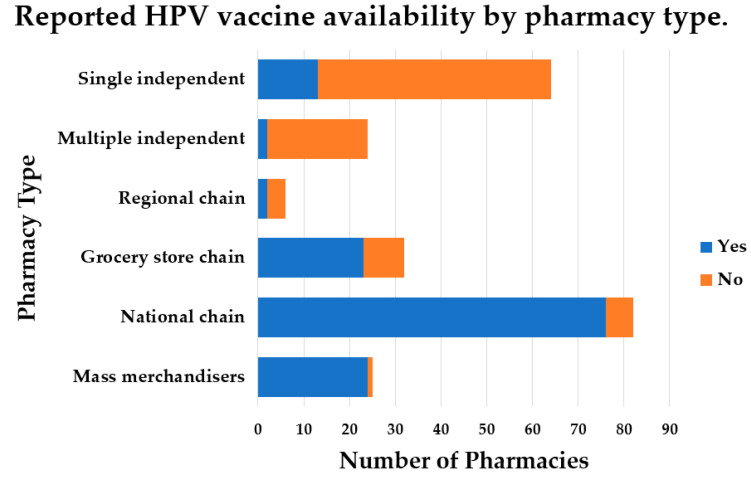
Reported HPV vaccine availability by pharmacy type. The total number of pharmacies, stratified by type, is indicated by each bar. Each bar is stacked to display the proportion of pharmacies, within that pharmacy type, that have the HPV vaccine available and those that do not have the HPV vaccine available. Chi-squared analysis revealed a significant difference in HPV vaccine availability across pharmacy type (*p* < 0.001).

**Figure 2 vaccines-10-00351-f002:**
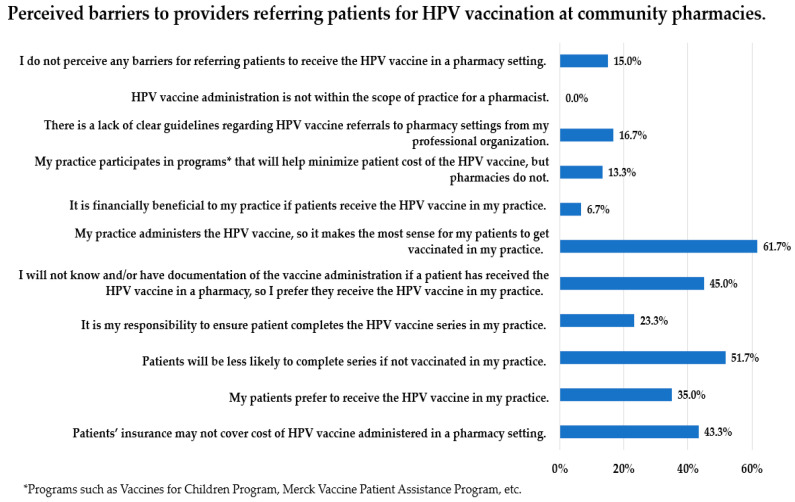
Perceived barriers to providers referring patients for HPV vaccination at community pharmacies. Each bar presents the percentage of respondents that indicated the barrier listed was a barrier for referring patients for HPV vaccination at a community pharmacy. The total percentage summed from all bars exceeds 100% because this survey question was a “mark all that apply” format question.

**Table 1 vaccines-10-00351-t001:** Pharmacy characteristics.

All Pharmacies (*n* = 233)	
Pharmacy Type	
Single independent	64 (27.5%)
Multiple independent	24 (10.3%)
Grocery store chain	32 (13.7%)
Mass merchandiser	25 (10.7%)
National chain	82 (35.2%)
Regional chain	6 (2.6%)
Geographic location	
Nonrural	171 (73.4%)
Rural	62 (26.6%)
HPV Vaccine Availability	
No	93 (39.9%)
Yes	140 (60.1%)
**Pharmacies with HPV Vaccine Availability (*n* = 140)**	
Stock HPV Vaccine	
No	46 (32.9%)
Yes	94 (67.1%)
Appointment Needed	N = 140
No	126 (90.0%)
Yes	14 (10.0%)
Length of Time Needed for Desired HPV Vaccination	
Same Day	107 (76.4%)
Within 24 h	9 (6.4%)
24–48 h	6 (4.3%)
More than 48 h	11 (7.9%)
Other	7 (5.0%)

**Table 2 vaccines-10-00351-t002:** Willingness of provider to refer an eligible patient for HPV vaccination in a community pharmacy, stratified by patient age group.

	Not Willing at All	Not Really Willing	Somewhat Willing	Willing	Very Willing
Patient Age Group	% (*n*)	% (*n*)	% (*n*)	% (*n*)	% (*n*)
11–18 years	0 (0)	0 (0)	5.2 (3)	25.9 (15)	68.9 (40)
19–26 years	0 (0)	0 (0)	8.3 (5)	25.0 (15)	66.7 (40)
27–49 years	0 (0)	0 (0)	10.0 (6)	23.3 (14)	66.7 (40)

## Data Availability

These data are not publicly available, but the data presented in this study are available upon reasonable request from the corresponding author.

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
