# Peer review of "Availability of the HPV Vaccine in Regional Pharmacies and Provider Perceptions Regarding HPV Vaccination in the Pharmacy Setting"

_vaccines, 2022, doi:10.3390/vaccines10030351_

Round 1

Reviewer 1 Report

Review for MDPI manuscript

The abstract looks good

The premise of the study is to assess the availability of the HPV vaccine in community pharmacies.  Additionally, the authors surveyed Family Medicine and OB/GYNs about their perceptions and practices referring patients to pharmacies to receive the HPV vaccine.

**It is interesting that you did not survey pediatricians.  There will  need to be a justification as to why you did not include pediatricians. Is this because more pediatricians stock these vaccines? 

Introduction:

Lines 36-8  -- Maybe include other cancers caused by HPV (anal) and in just men (penile) and women (cervical, vaginal)

Line 54 – change vaccination to vaccine

You mention the disparities in HPV vaccine uptake in rural regions.  Somewhere it would be interesting to see how much of your population is considered rural.

It is interesting that your team called the pharmacies asking as a secret shopper.  Could Merck have given you the list of pharmacies that stock the vaccine? This is merely an observation – not something you need to address for a revision.

Results

You had a great response rate.

The information under Table 1 – Should this be put in the main text of the document?  This is interesting information. I was looking for the information about differences between rural and urban pharmacies – and found that under Table 1.  However, that seems like, based on how you framed the introduction, that information deserves a little bit more space.

The healthcare providers you surveyed – Did any of them practice in more rural regions.  I find that folks who are also affiliated with a university are much more willing to complete a survey or participate in research.  Are they representative of other practicing physicians? If so, could you add 1 sentence in your limitations (after you list that your providers came largely from the university healthcare system) about how this could impact your results?

Line 17: Remove 1 “Most” (there are 2)

That is interesting that 98% said they administer the vaccine and 95% say that their practice stocks the vaccine. I would imagine these %s would be the same.

I wonder what made providers want to prescribe getting the vaccine elsewhere (health department) if they stocked the vaccine.  It would make sense that they recommend receiving the booster doses for HPV at another location (due to convenience).  Did they refer out for that first dose? This may be worth noting in the Discussion.

This group is willing to refer patients to pharmacies.  Good!  It may be because they are used to working with adult populations instead of children. They may refer adults to get vaccines such as the flu and shingles vaccines more regularly – making them more comfortable recommending this vaccine to adolescent, young adults, and adults.  ***This would be something interesting to note in your discussion – how your healthcare providers differed from past healthcare providers (namely pediatricians) in regards to their willingness to refer patients to community pharmacies for vaccines.

The “pharmacy benefit” is a real problem for many insurances – meaning, that health insurance plans will not cover the vaccine if it is not administered by a physician, nurse, etc.  Patients would be expected to pay out of pocket for this expensive vaccine series.

Line 268 – By law, pharmacists have to fax a Vaccine Information Sheet to providers.  However, whether patients have accurate information about their PCP/PCP fax machine (and who is their PCP) and whether the clinics/offices see this fax and file the VIS is another question.

Reviewer 2 Report

School-based vaccinations and community education increased the HPV vaccination rate at a greater rate than education only (Kaul 2019). For people who missed the HPV vaccine offered in school, a high availability of the HPV vaccine is important. HPV vaccination in the pharmacy setting may increase the coverage of the HPV-vaccine. Maples et al. have studied the HPV vaccine availability in community pharmacies and the willingness of and perceived barriers to referring patients for HPV vaccination in a pharmacy setting. Results indicated the HPV vaccine was available in most pharmacies. Providers were willing to refer patients to a community pharmacy for HPV vaccination, despite this not being a common practice, likely due to numerous barriers reported.

The claims are properly placed in the context of the previous literature. The experimental data support the claims. The manuscript is written clearly enough that most of it is understandable to non-specialists. The authors have provided adequate proof for their claims, without overselling them. The authors have treated the previous literature fairly. The paper offers enough details of methodology so that the experiments could be reproduced.

Minor revision

Line 176, “Most Most respondents indicated their practice administered” => “Most respondents indicated their practice administered”

References

Sapna Kaul, Thuy Quynh N. Do, Enshuo Hsu, Kathleen M. Schmeler, Jane R. Montealegre, Ana M. Rodriguez. School-based human papillomavirus vaccination program for increasing vaccine uptake in an underserved area in Texas, Papillomavirus Research, Volume 8, 2019, 100189, ISSN 2405-8521, https://doi.org/10.1016/j.pvr.2019.100189.

(https://www.sciencedirect.com/science/article/pii/S2405852119300114)
